# Cost–Benefit Evaluation of an Organizational-Level Intervention Program for Decreasing Sickness Absence among Public Sector Employees in Sweden

**DOI:** 10.3390/ijerph19052998

**Published:** 2022-03-04

**Authors:** Jonathan Severin, Mikael Svensson, Magnus Akerstrom

**Affiliations:** 1Region Västra Götaland, The Institute of Stress Medicine, 413 19 Gothenburg, Sweden; magnus.akerstrom@vgregion.se; 2School of Public Health and Community Medicine, Institute of Medicine, Sahlgrenska Academy, University of Gothenburg, 413 90 Gothenburg, Sweden; mikael.svensson.2@gu.se

**Keywords:** absenteeism, health economic evaluation, occupational health and safety, organizational, workplace

## Abstract

Work-related illnesses create a vast economic burden for employers and society. Organizational-level workplace interventions are recommended to prevent these illnesses, but the knowledge about the economic benefits of such interventions is scarce. The study aimed to evaluate the economic benefit of an organizational-level workplace program for decreasing sickness absence. The program contained a monetary support approach (MSA) and an approach combining monetary and facilitator support (FSA). Cost–benefit analyses were used, where the results were compared to those of business as usual. Economic benefits of reduced sickness absence were based on the value of reduced production loss and direct sick pay costs, respectively. Sensitivity analyses were used to assess the robustness of the results. The program had a positive net benefit when measuring productivity loss, where the FSA had a net benefit and the MSA had a net loss. A negative net benefit was derived when measuring direct sick pay costs. The intervention effect on sickness absence affected the net benefit the most. This program was economically beneficial in terms of reducing the productivity loss, but not of reducing direct sick pay costs connected to short-term sickness absence. Using evidence-based methods is essential for increasing the long-term net benefit of organizational-level workplace interventions.

## 1. Introduction

Work-related illness and increasing rates of sickness absence are a growing concern, especially in the public sector [1,2]. Other than musculoskeletal problems, mental health disorders are the most common reason for sickness absence in Sweden [3], and in many cases, a dysfunctional work environment with unsatisfactory working conditions contributes to work-related illnesses [4]. Alongside the increased sickness absence, adverse working conditions also increase presenteeism (i.e., working with reduced productivity because of illness), employee turnover, and productivity loss, which creates a vast economic burden, both to the employer and to society [5,6]. In Sweden in 2018, the societal cost for total sickness absence was estimated to be SEK 64 billion (approx. EUR 6.4 billion), the value of production loss due to sickness absence being the largest cost component [7].

The workplace is an important arena for reducing work-related illness, and workplace interventions are often implemented to prevent accidents, improve the work environment, improve employee health, and reduce sickness absence [4,8]. Since economic resources for occupational health are scarce, resources should be allocated to workplace interventions that are effective in terms of both improving employee health and generating economic benefits [9]. Individual, group, and/or organizational-level interventions can be used [10,11]. Interventions on an individual level aim to improve employees’ physical or mental health, often by using lifestyle activities or stress management [8,11], while group-level interventions target the interpersonal interaction between employees, focusing on the social work environment [12]. Lastly, interventions on an organizational level aim to improve working conditions by focusing on the workplace structure, policies, procedures, routines, and management practices, rather than to strengthen individual employees [12,13,14]. Even though organizational-level interventions, or interventions administered on a combination of levels, have been recommended because they can achieve long-term positive effects on working conditions [14,15], most interventions still use individual-level approaches [12]. Therefore, there is limited evidence and information on how organizational-level interventions should be designed and executed in practice [13,16]. Furthermore, systematic reviews evaluating the economic benefits of workplace interventions have shown varying results, where some studies show positive benefits, while others report negative results [17,18,19]. Therefore, there is a gap in knowledge regarding economic benefits of workplace interventions. More high-quality studies are needed [19], especially regarding organizational-level interventions [20]. Such knowledge can create incentives for the employer and help decision makers implement evidence-based interventions that are effective, both when it comes to reducing work-related illness and in terms of the economic burden of these illnesses.

In 2017, an organizational-level workplace program with the purpose to decrease sickness absence was implemented in a public sector organization in Sweden, using two different approaches. The implementation process and the effects of this program have previously been evaluated, showing differences in effects on sickness absence for the two approaches [21,22,23]. The aim of the current study is to conduct an economic evaluation comparing the costs and benefits of this program. The objectives are to evaluate the costs and benefits of the program compared to business as usual (i.e., when managers perform their ordinary work environment management), using sickness absence to assess both reduced productivity loss and direct sick pay costs connected to short-term sickness absence (≤14 days). The results are then used to compare the costs and benefits between the two approaches in order to increase the knowledge about the economic benefits of the different designs.

## 2. Materials and Methods

### 2.1. Setting and Program Design

The present study was carried out in Region Västra Götaland, a public sector employer in Sweden with approximately 55,000 employees. About 85% of the employees work in the health care sector; the rest work in culture, education, public transport, and regional development.

In 2017 and 2018, a program was implemented as part of a regional political initiative, with the purpose of improving working conditions and reducing sickness absence for the employees. A total of SEK 15 million (MSEK 15) (EUR 1.5 million) was allocated annually. Two different approaches were used in the program: one consisting of monetary support, and the other consisting of both monetary and facilitator support (Figure 1). In the monetary support approach (MSA), line managers together with Human Resources (HR) were given the opportunity to apply for monetary support to design and implement interventions at their workplace. In total, 154 applications (including 209 interventions) for monetary support were submitted, of which 86 applications (107 interventions) were approved and implemented. After excluding applications where the workplace could not be matched with data from the region’s employee system, 71 intervention groups were eligible for evaluation. The majority of the intervention groups worked with patient care (93%, *n* = 66), and the remaining with service functions within health care (7%, *n* = 5). Almost exclusively, applications described organizational-level work environment challenges, such as staff shortages, high workload, and unclear goals or tasks. Interventions of all types were implemented, including workshops, physical activity, team building, work environmental analyses, manager support, and structural changes [22].

In the monetary and facilitator support approach (FSA), eight operational areas with a combination of high total sickness absence (>10%) and high employee turnover were invited to participate in the program. As it was not possible to include all workplaces in the operational areas, one workplace from each area was selected. In total, 88% (*n* = 7) of the workplaces worked with patient care, and the remaining with service functions within health care (12%, *n* = 1). External process facilitators (EPFs) with knowledge in work environment and change management were then assigned to each workplace, and a strategic group, comprising managers and HR, was formed as part of the intervention. During 2017 and 2018, the strategic group, with support from the EPFs, identified work environment challenges and implemented organizational-level interventions, e.g., analysis of organizational conditions, management coaching, and workshops on work conditions, culture, and roles and responsibility. All interventions in this approach were considered to fit the context and challenges of the workplace [23]. In both approaches, the interventions were implemented between early 2017 and late 2018; the mean implementation start was in January 2018.

### 2.2. Estimating Intervention Effects on Sickness Absence

To estimate intervention effects, monthly data on total and short-term (≤14 days) sickness absence (vacations, parental leave, and caring for sick children deducted) was collected from the region’s administrative employee system. Data was collected between January 2016 and March 2020 for the MSA, and between January 2015 and October 2019 for the FSA, as the different effect evaluations were performed separately. To separate the intervention effects from effects due to other concurrent changes at the workplaces, data was also collected from reference groups within the same operational areas with similar context and challenges as the intervention groups, as matching control groups could not be found in the region. The region consists roughly of three hierarchical levels: departments, operational areas, and units; the reference groups were constructed from the aggregated means of the operational area (e.g., clinic) of which the intervention group (e.g., surgery unit) was a part (intervention group excluded). The intervention effect on total and short-term sickness absence per month has previously been evaluated for the two approaches [21,23]. In this study, the combined effect of the MSA + FSA was evaluated (Table 1). To be able to control for time trends, seasonality, and autocorrelation, mixed-effect models were used separately for the intervention and reference groups, as data was only available on a higher aggregated level for the reference groups (the average sickness absence for the respective operational area).

To be able to assess benefits, an overall intervention effect had to be calculated, which was done first by subtracting the estimated effect for the reference group from the estimated effect for the respective intervention group. Thereafter, 95% confidence intervals (CIs) were calculated by summing the variances of the different groups, for total and short-term sickness absence, respectively. The standard error (SE) was calculated as the square root of the summed variance, and lastly, SE of the difference was used to calculate the 95% CI. SAS version 9.4 software (SAS Institute, Cary, NC, USA) and SPSS Statistics version 25 (IBM, Armonk, New York, NY, USA) were used for statistical analyses, using two-sided CIs with significance determined at *p* < 0.05.

### 2.3. Identifying and Measuring Costs

Costs were collected by invoices and included costs for facilities, occupational health care services, external consultants, and facilitator support. For facilitator support, the costs were estimated by multiplying the number of hours used by the EPFs (5000 h each year) by the hourly wage for the EPFs including payroll taxes and fixed overhead costs during 2017 (SEK 289/h) and 2018 (SEK 306/h).

### 2.4. Identifying and Measuring Benefits

Benefits were measured as the cost saving from the intervention’s effect of reducing sickness absence, calculated as the monthly difference in hours of sickness absence before and after the intervention, i.e., benefits per month = hours of sickness absence before the intervention—hours of sickness absence after the intervention. The hours of sickness absence before the intervention were calculated as the mean hours of sickness absence per month prior to the intervention for the intervention groups. The hours of sickness absence after the intervention for the intervention groups were calculated as the mean hours of sickness absence per month post-intervention. Because the effect evaluations were performed separately and therefore included a shorter time horizon for the FSA, benefits were limited to 34 months (January 2017 to October 2019), to be able to compare the costs and benefits between the two approaches.

### 2.5. Cost–Benefit and Sensitivity Analyses

The cost–benefit analyses (CBA) were performed based on two different measures. In the first, reduced productivity loss was measured by including benefits from reduced total sickness. The human capital approach was used, meaning that wages were used as the value of productivity loss, i.e., each hour not worked was equal to 1 h of productivity loss [9]. Benefits were therefore multiplied by the intervention group’s mean hourly wage for the year 2018 (SEK 309/h including payroll taxes and fixed overhead costs).

In the second measure, reduced direct sick pay costs were measured by including benefits from reduced short-term sickness absence (≤14 days). In Sweden, 80% of the wages, apart from 1 qualifying day, are disbursed by the employer as sick pay during the first 14 days of absence. After the initial 14 days of absence, the responsibility for paying sick pay to the individual is transferred to the government [24]. Therefore, direct sick pay costs were estimated to be 80% of the benefits of the short-term sickness absence (also calculated based on the mean wage in 2018), excluding 1/14 of the cost because of the qualifying day. All costs and benefits were discounted 3%, according to Swedish recommendations [25], adjusted for inflation and expressed at 2020 price level. Lastly, costs were subtracted from benefits to assess the net present value (NPV), where an NPV > 0 meant that the interventions were economically beneficial. Net benefit per year, cost–benefit ratio, and benefits per employee per year were also calculated.

Deterministic and probabilistic sensitivity analyses were conducted in two steps to test the robustness of the results. In the first step, number of employees, wages, and intervention costs were increased/decreased separately by 30%, while intervention effect was varied by using the lower/upper estimates in the 95% CI. In the second step, 1000 iterations were randomly assessed, simultaneously changing the same variables, using gamma distribution for intervention costs, normal distribution for effects, and uniform distribution for number of employees and wages (random change between 30% increase/decrease). The percentage of iterations with an NPV > SEK 0 was calculated. Microsoft Excel 365 version 2008 (Microsoft Corporation, Seattle, WA, USA) was used for the CBA and sensitivity analyses.

## 3. Results

The total cost for the program was approximately MSEK 10.5 (approx. EUR 1.1 million) (Table 2). The cost per employee was higher for the FSA compared to that for the MSA (SEK 3410/employee vs. SEK 1560/employee). An overall decrease in sickness absence was seen due to the program (–0.31 percentage points, 95% CI –0.88 to 0.25 in total sickness absence; –0.08 percentage points, 95% CI –0.28 to 0.11 in short-term sickness absence), but the effect was only significant for total sickness absence with the FSA (–1.9 percentage points, 95% CI –2.99 to –0.87) (Table 2).

### 3.1. Net Benefits for the Program and the Two Approaches

In the CBA based on productivity loss, the NPV per year was approximately MSEK 1.7 for the program, MSEK −0.2 for the MSA, and MSEK 9.6 for the FSA. The FSA also had a much larger net benefit per employee per year (SEK 5850/employee) compared to that of the MSA (SEK −60/employee). In the CBA based on sick pay, a negative NPV was seen for the program and the two approaches. The net loss per year was almost equal for the MSA and the FSA (MSEK −1.2), although the net loss per employee per year was largest for the FSA (SEK −720/employee, compared to SEK −370/employee for the MSA) (Table 3).

### 3.2. Sensitivity Analyses of the Results

In the deterministic sensitivity analyses, the result was most sensitive to intervention effects, somewhat sensitive to intervention costs and wages (in the MSA), but robust for number of employees (Figure 2). In the probabilistic sensitivity analyses (Figure 3), 58% of the iterations for the program in the CBA based on productivity loss had a positive NPV (47% for the MSA and 100% for the FSA). For the CBA based on sick pay, 2% of the iterations for the program had a positive NPV (10% for the MSA and 6% for the FSA, respectively).

## 4. Discussion

In this study, the costs and benefits of an organizational-level workplace program were evaluated. Differences in costs and benefits were seen between the two approaches used in the program, i.e., monetary support (MSA) versus monetary and facilitator support (FSA). Differences were also seen between the methods used for measuring benefits.

When comparing the approaches, the cost per employee was more than twice as high for the FSA. Nevertheless, a net benefit was seen with this approach, while a net loss was seen for the MSA. This difference in economic benefits between the approaches could be explained by the higher intervention effect in the FSA compared to that in the MSA, where the process evaluation of the two approaches showed that line managers and HR using the MSA needed support with designing interventions that corresponded to the workplace challenges they were facing. Only half of the interventions in the MSA were considered to fit the workplace challenges [22]; by contrast, there was a high correspondence between implemented interventions and the workplace challenges for the FSA [21]. The importance of fitting interventions to the context and workplace challenges has been addressed in previous literature [26,27]. Therefore, improving the fit between interventions and workplace challenges could increase the economic benefit even further. This can be done by using EPFs or other ways of supporting managers in designing effective interventions. Consequently, decision makers should not fear high initial costs if they use evidence-based approaches, as also shown in the deterministic sensitivity analyses, where increased intervention costs in the FSA would still have produced a net benefit.

Another way of improving the economic benefit is to improve the implementation of the interventions. When assessing the presence of supporting factors (e.g., commitment or knowledge about the organizational work environment) in a previous evaluation of the FSA, it was seen that workplaces with a greater number or range of supporting factors in place had a greater decrease in sickness absence than workplaces without these factors did [28]. Positive economic results have also been seen for workplace interventions where supporting factors were present, such as strong support from managers, and high participation [29].

There was, however, a difference in sickness absence between the participating workplaces with lower total sickness absence in the MSA, suggesting that the potential for impacts was smaller. Interestingly, a previous evaluation showed that the intervention groups with this approach had a significantly higher sickness absence than their respective reference groups, and a higher sickness absence compared to the mean sickness absence in the region [22], implying a potential to reduce sickness absence for workplaces in both approaches.

The net benefit was larger for the CBA based on productivity loss, compared to the CBA based on direct sick pay costs. This was expected, as the former was based on total sickness absence, while the latter was based on short-term sickness absence only. From an employer/payer perspective, a net loss regarding direct sick pay costs could be interpreted to mean that the program is not cost-worthy; however, benefits from reduced productivity loss could also be of relevance. Previous research has shown that presenteeism contributes more to productivity loss than does absenteeism [6]. In a Swedish survey, one out of three participants reported presenteeism two or more times during the last year, where the highest amount of presenteeism was found in the health care, welfare, and education sectors [30]. The present program was implemented mostly in the health care sector, where presenteeism could be contributing substantially to the total cost for the employer. Therefore, the total benefits of the program could have been underestimated, especially when using only direct sick pay costs.

In addition, the aim of this program was to reduce total sickness absence, rather than short-term sickness absence. Some short-term sickness absence could even be favorable. Dellve et al. (2011) have concluded that balanced work attendance (at most from four to seven days of sick leave per employee per year) was associated with improved health and performance; by contrast, sickness attendance was associated with poor health, burnout, and sickness absence [31]. When taking these factors into account, the incentives for the employer/payer to implement organizational-level workplace interventions may be increased, as a reduction in costs other than sick pay costs or productivity loss could produce economic benefits. Moreover, this program included monetary support, which could be used as a policy tool to improve occupational health [32]. Consequently, implementing organizational-level interventions using evidence-based approaches with additional funds may be effective not only in reducing sickness absence, but also in producing economic benefits.

### Strengths and Limitations

In this study, one limitation is that reference groups within the same operational areas were used to separate the intervention effect and for comparison with “business as usual.” One uncertainty using this method is that other interventions could have been implemented in the reference groups during the time of the intervention; however, it was not possible for any workplaces to participate in both approaches. It was also assumed that the difference in effect on sickness absence was caused by the implemented approach, even though workplaces that participated in the different approaches might differ in terms of context and workplace challenges. However, the program was implemented within the same organization, where workplaces mostly in the health care sector participated. Additionally, participation was based on having high rates of sickness absence and work environment challenges [22].

Another limitation is that the human capital approach was used in the CBA, where benefits could have been overestimated, as productivity loss because of sickness absence could be compensated on return to work, or by colleagues [9]. Additionally, only direct costs for the interventions were included, but not costs for time used by managers and employees to participate in the interventions, where it can be argued that managers have a responsibility to spend time to manage the work environment. This was also seen in a previous survey, where decision makers considered such costs to be part of their daily responsibility [33].

A third limitation is that, due to limitations in the administrative system, it was only possible to retrieve aggregated data on a higher hierarchical level for the reference groups. One strength is that aggregated register data on sickness absence were used to evaluate costs and benefits, as such data are of higher quality than self-assessed data.

## 5. Conclusions

This program led to reduced productivity loss, where the net benefit was larger for the approach with both monetary and facilitator support, despite larger intervention costs, compared to the approach where managers designed and implemented interventions with monetary support but had no support from a facilitator. The intervention effect on sickness absence was the most important factor to produce a net benefit. It is important to implement interventions that fit the workplace challenges in order to achieve economic benefits from organizational-level workplace interventions, and external or other support can be used to help managers choose effective interventions. When evaluating reduced direct sick pay costs, this program produced a net loss, although benefits could have been underestimated, as the inclusion of other costs, such as the reduction in presenteeism and employee turnover, as well as time of return to work or rehabilitation costs, could affect the benefits to the employer. Therefore, future economic evaluations should also consider aspects other than reduced productivity loss and sick pay. In tackling the issue of high sickness absence in the public sector, evidence-based approaches with additional funds could be important tools to employers and decision makers when implementing organizational-level workplace interventions.

## Figures and Tables

**Figure 1 ijerph-19-02998-f001:**
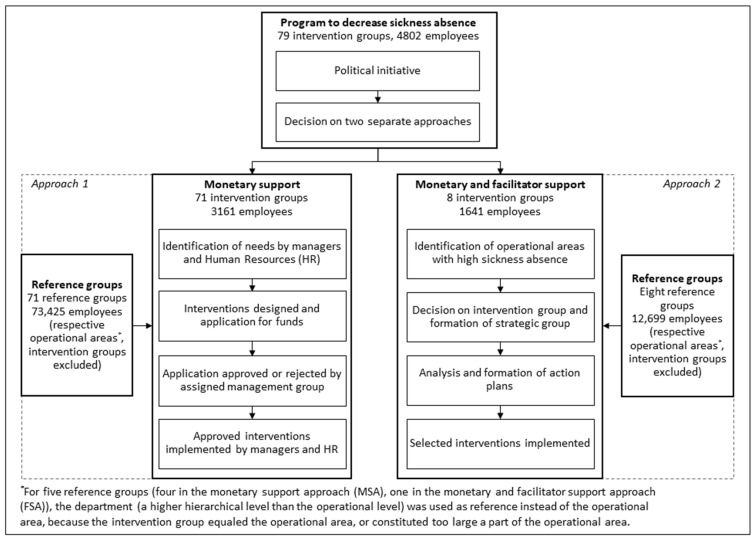
Flowchart describing the different steps and groups in the program.

**Figure 2 ijerph-19-02998-f002:**
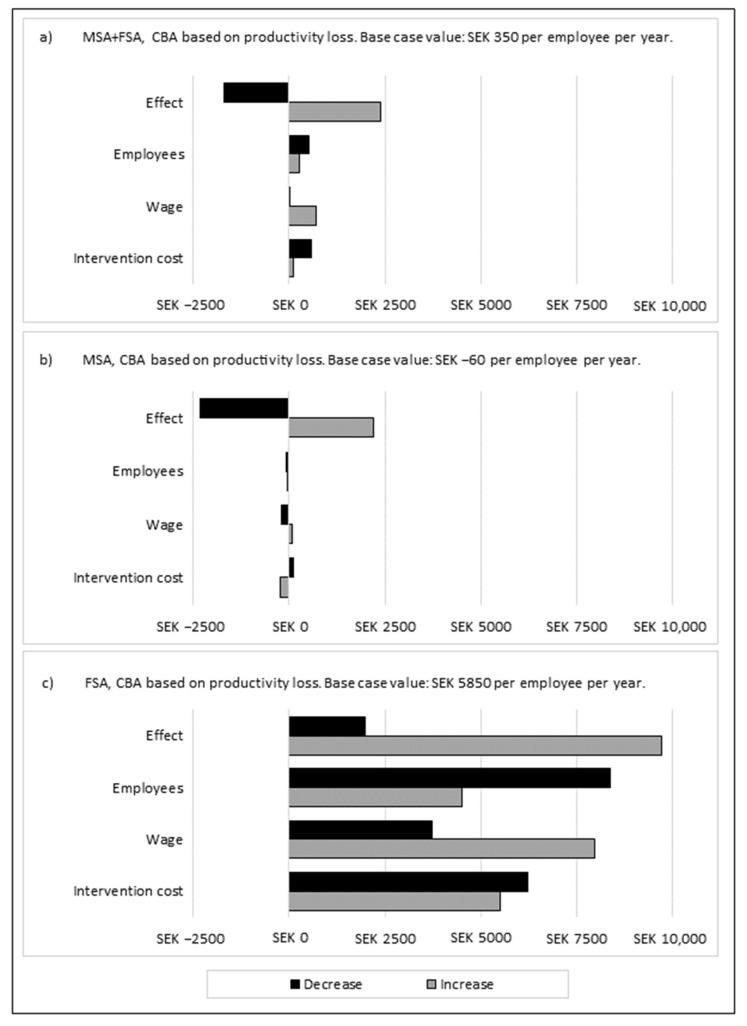
Deterministic sensitivity analysis for the CBA based on productivity loss for the (**a**) MSA + FSA, (**b**) MSA, (**c**) FSA, and for the CBA based on sick pay for the (**d**) MSA + FSA, (**e**) MSA, and (**f**) FSA. Number of employees, wages, and intervention costs decreased and increased by 30%. Effects changed using the upper and lower interval of the 95% confidence interval (CI) for the estimated intervention effect. All costs are expressed at 2020 price level. CBA = cost–benefit analysis; FSA = monetary and facilitator support approach; MSA = monetary support approach.

**Figure 3 ijerph-19-02998-f003:**
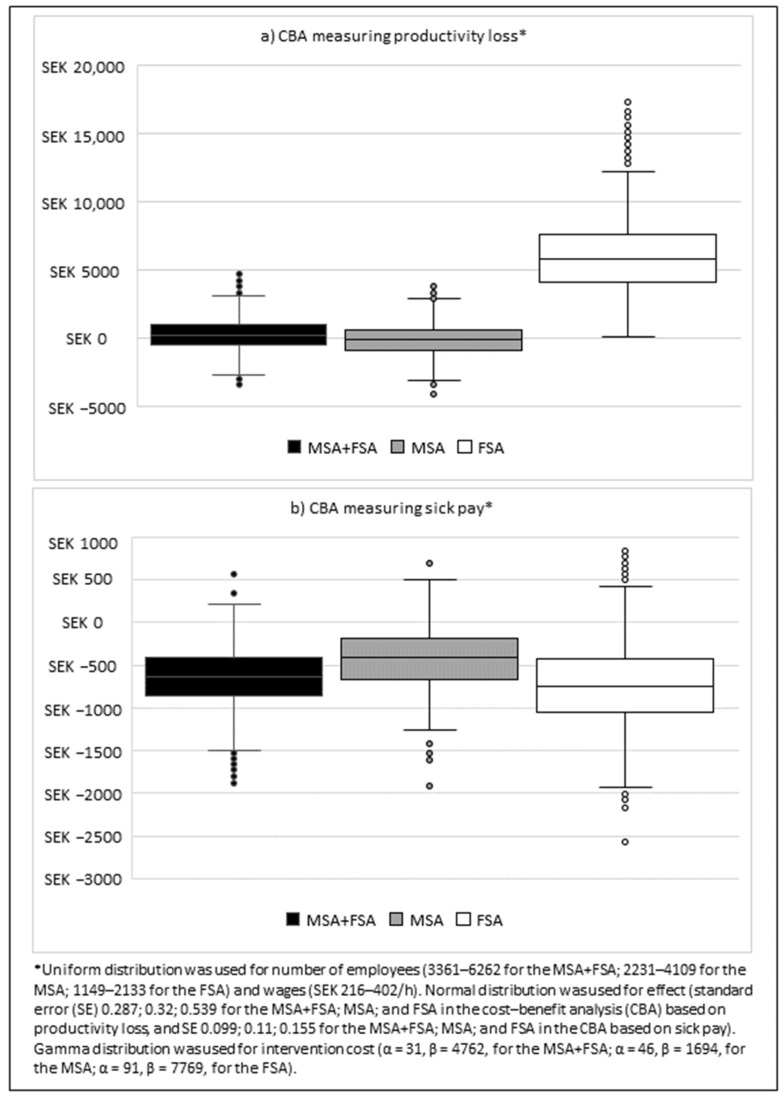
Probabilistic sensitivity analysis using 1000 iterations for the monetary support approach (MSA), the monetary and facilitator support approach (FSA), and MSA + FSA together, giving the percentage of iterations with positive cost benefits, for (**a**) the CBA measuring productivity loss, and (**b**) the CBA measuring sick pay.

**Table 1 ijerph-19-02998-t001:** Description of the intervention groups, reference groups, and estimated effect of the program on sickness absence.

		MSA + FSA		Monetary Support Approach (MSA)		Monetary and Facilitator Support Approach (FSA)	
		Intervention groups (workplaces) *n* = 79	Reference groups (operational areas *) *n* = 79	Intervention groups (workplaces) *n* = 71	Reference groups (operational areas *) *n* = 71	Intervention groups (workplaces) *n* = 8	Reference groups (operational areas *) *n* = 8
Employees per group (*n*)	*Mean* *(range)*	61 (11–458)	1090 (23–15,435)	45 (11–191)	1034 (23–15,435)	205 (41–458)	1587 (113–3179)
Total sickness absence per month, pre-intervention (%) ^†^	*Mean* *(range)*	8.5 (1.0–17.8)	7.1 (1.2–12.7)	8.2 (1.0–17.8)	7.0 (1.2–12.7)	11.3 (9.1–14.1)	7.6 (6.1–9.8)
Short-term sickness absence (≤14 days) per month, pre-intervention (%) ^†^	*Mean* *(range)*	3.0 (0.4–5.4)	2.5 (0.8–3.7)	2.9 (0.4–5.4)	2.5 (0.8–3.7)	3.5 (2.5–4.8)	2.6 (1.9–3.1)
Estimated effect on total sickness absence (percentage points)	*β* *(variance)*	−0.56 (0.067)	−0.25 (0.016)	−0.43 ^‡^ (0.083)	−0.29 ^‡^ (0.019)	−1.9 ^‡^ (0.247)	0.03 ^‡^ (0.044)
Estimated effect on short-term sickness absence ≤ 14 days (percentage points)	*β* *(variance)*	−0.18 (0.008)	−0.1 (0.002)	−0.2 ^‡^ (0.009)	−0.13 ^‡^ (0.003)	−0.11 ^‡^ (0.02)	0.07 ^‡^ (0.004)

** *For five reference groups (four in the MSA, one in the FSA), the department (a higher hierarchical level than the operational level) was used as reference group instead of the operational area, because the intervention group equaled the operational area or constituted too large a part of the operational area. ^†^ As the intervention started at different times for different groups, the number of included months ranged from 12 to 44 months. ^‡^ Previously published data [21,23].

**Table 2 ijerph-19-02998-t002:** Total costs (without discounting) and estimated intervention effect on sickness absence for the different approaches; 95% CI = 95% confidence interval; FSA = monetary and facilitator support approach; MSA = monetary support approach.

	MSA + FSA	MSA	FSA
**Costs collected through invoices**			
2017 (MSEK)	2.438	2.260	0.178
2018 (MSEK)	5.124	2.683	2.441
**Costs for facilitator support**			
2017 (MSEK)	1.444	0	1.444
2018 (MSEK)	1.532	0	1.532
Total direct costs (MSEK)	10.538	4.943	5.594
Cost per employee (SEK)	2200	1560	3410
**Estimated intervention effect ***			
Total sickness absence (percentage points)	−0.31	−0.14	−1.93 ^†^
(95% CI)	(−0.88–0.25)	(−0.77–0.49)	(−2.99–−0.87)
Short-term sickness absence (percentage points)	−0.08	−0.07	−0.18
(95% CI)	(−0.28–0.11)	(−0.29–0.15)	(−0.48–0.12)

* The estimated overall intervention effect in sickness absence between the intervention and reference groups, calculated by subtracting the effect for the reference groups from the intervention groups. ^†^ Statistically significant.

**Table 3 ijerph-19-02998-t003:** Cost–benefit analysis (CBA) adjusted for inflation and expressed at 2020 price level, with costs and benefits discounted 3%. FSA = monetary and facilitator support approach; MSA = monetary support approach.

	Total Net Benefit (MSEK) *	Net Benefit per Year (MSEK)	Cost–Benefit Ratio	Net Benefit per Employee per Year (SEK)
CBA based on productivity loss ^†^				
MSA + FSA	4.796	1.693	1.45	350
- MSA	−0.523	−0.184	0.90	−60
- FSA	27.213	9.605	5.83	5850
CBA based on sick pay ^†^				
MSA + FSA	−7.615	−2.688	0.29	−560
- MSA	−3.351	−1.183	0.33	−370
- FSA	−3.361	−1.186	0.40	−720

* Includes benefits from January 2017 to October 2019, i.e., 34 months. ^†^ The CBA based on productivity loss used total sickness absence, while the CBA based on sick pay used short-term sickness absence.

## Data Availability

The datasets used and/or analyzed during the current study are available from the corresponding author on reasonable request.

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
