# Peer review of "Cost–Benefit Evaluation of an Organizational-Level Intervention Program for Decreasing Sickness Absence among Public Sector Employees in Sweden"

_ijerph, 2022, doi:10.3390/ijerph19052998_

Round 1

Reviewer 1 Report

The authors performed cost-benefit analyses of an organizational-level intervention program for decreasing absence among public sector employees in Sweden. The topic of the study is relevant to International Journal of Environmental Research and Public Health. The study was well planned and executed. The manuscript was well written. It should be publishable after minor corrections.

On page 1, line 13: “The aim was to …” should be written as “The study aimed to …”

On page 8, line 203: “-3.0 to 0.87” should be “-2.99 to -0.87” (see Table 2).

On page 8, Table 2: Please present the 95% confidence intervals appropriately i.e. using “en dashes” rather than “x’s”.

On page 15, lines 376-377: The publisher of Hassard et al.’s (2014) report should be “European Agency for Safety and Health at Work.”

On page 15, lines 383-384: The authors should be “Bauer, G.; Davis, J.K.; Pelikan, J.” Please delete “Group, o.b.o.t.E.T.W., Constortium, T.E.” between “Bauer, G.” and “Davis, J.K.”

On page 16, lines 394-396. The year of this publication should be “2015.”

On page 16, lines 415-417. Please delete one of “10.1007/s00420-021-01686-y”.

On page 16, line 418. The full title of this report should be “How to Reduce Sickness Absences in Sweden: Lessons from International Experience. Details should include “OECD Economics Department Working Papers No. 442, OECD, Paris.”

Author Response

Thank you for this positive feedback and for reviewing our manuscript and the valuable feedback on the study and the manuscript. Please see comments for corrections done below, highlighted in red.

On page 1, line 13: “The aim was to …” should be written as “The study aimed to …”
This has been changed according to suggestion.

On page 8, line 203: “-3.0 to 0.87” should be “-2.99 to -0.87” (see Table 2).
Thank you for noticing this rounding error. This is now changed.

On page 8, Table 2: Please present the 95% confidence intervals appropriately i.e. using “en dashes” rather than “x’s”.
The x’s are now changed to “–“.

On page 15, lines 376-377: The publisher of Hassard et al.’s (2014) report should be “European Agency for Safety and Health at Work.”
The publisher is now added to the reference.

On page 15, lines 383-384: The authors should be “Bauer, G.; Davis, J.K.; Pelikan, J.” Please delete “Group, o.b.o.t.E.T.W., Constortium, T.E.” between “Bauer, G.” and “Davis, J.K.”
“Group…” is now deleted between the two other authors.

On page 16, lines 394-396. The year of this publication should be “2015.”
The year 2015 of the publication is now added

On page 16, lines 415-417. Please delete one of “10.1007/s00420-021-01686-y”.
One of the DOI’s is now deleted.

On page 16, line 418. The full title of this report should be “How to Reduce Sickness Absences in Sweden: Lessons from International Experience. Details should include “OECD Economics Department Working Papers No. 442, OECD, Paris.”
This reference is now changed according to suggestion.

Reviewer 2 Report

I really liked the way this paper was written. The authors make a very thorough  analysis of conducting a cost-benefit analysis of two programs implemented at certain organizations. These programs, called MSA and FSA, were adopted by certain establishments, with the reference group comprising of companies that went with a "business as usual" format. The authors clearly explained the way they conduct the cost benefit analysis in  the paper. They find that FSA has a stronger impact and has a net benefit in reducing productivity loss, with negative net benefit for sick pay leave.

I have some minor comments that the authors might consider using to make their paper appeal to a broader audience. 

  1. I would like to know more information of the reference group that was used. I am not sure if the authors are comparing apples to apples based on how the reference group is described. I'm guessing that since majority of the participants were from health care sector, the reference group would be part of that industry. However, the health  care sector is vast. More detail on characteristics of  the reference group would help readers understand the context better.
  2. What type of work do the employees do. Productivity loss might be very different when a physician gets sick and takes sick leave compared to a cleaning staff. Both output and value of the output would be different. 
  3. Since most of the participants are from health care, I would consider dropping the others from the analysis and repeat the process. That might be another robustness check and might cross out the possibility that  most of the results are being driven by industry specific factors. 
  4. Please place *** (significance stars) next to the results that are statistically significant in the tables.  

Author Response

Thank you for this positive feedback and for reviewing our manuscript, and the valuable feedback on the study and the manuscript. We are glad to hear that you found the cost-benefit analysis clearly explained. Please see answers to your comments below, highlighted in red.

I have some minor comments that the authors might consider using to make their paper appeal to a broader audience. 

1. I would like to know more information of the reference group that was used. I am not sure if the authors are comparing apples to apples based on how the reference group is described. I'm guessing that since majority of the participants were from health care sector, the reference group would be part of that industry. However, the health  care sector is vast. More detail on characteristics of  the reference group would help readers understand the context better.

Thank you for highlighting this question of comparing the intervention and reference group. Unfortunately, matched controls could not be found within the organisation. Therefore, the reference groups constituted of workplaces within the same operational areas as the respective intervention group to make sure that they share the same context and challenges as the intervention groups. For example, one intervention group was the orthopaedic surgery within the orthopaedic clinic, where the entire orthopaedic clinic (intervention group excluded) was the reference group.

To make this clearer, we have made an addition to the section 2.2. Estimating Intervention Effects on Sickness Absence (p.5) in the methods:

“To separate the intervention effects from effects due to other concurrent changes at the workplaces, data was also collected from reference groups within the same operational areas with similar context and challenges as the intervention groups, as matching control groups could not be found in the region.”

2. What type of work do the employees do. Productivity loss might be very different when a physician gets sick and takes sick leave compared to a cleaning staff. Both output and value of the output would be different. 

Thank you for this comment. Unfortunately, we could only collect data on sickness absence aggregated on a workplace level, due to limitations in the employer’s administrative personnel system, and not individual employee data with occupation. However, it was possible to retrieve the targeted sector for the intervention groups.

To make this clearer, this is now added in the method section (2.1. Setting and Program Design), both for MSA and FSA:

MSA: The majority of the intervention groups worked with patient care (93%, n=66), and the remaining with service functions within healthcare (7%, n=5).

FSA: In total, 88% (n=7) of the workplaces worked with patient care, and the remaining with service functions within healthcare (12%, n=1).

3. Since most of the participants are from health care, I would consider dropping the others from the analysis and repeat the process. That might be another robustness check and might cross out the possibility that most of the results are being driven by industry specific factors. 

Thank you for this suggestion on another robustness check. As all participating workplaces belongs to the same employer, and in some way was connected to the healthcare sector (see answer to comment 2), we decided to keep all the participants in the analysis. But it is an interesting idea to separate different sectors to see if the results differ/are driven by sector specific factors, which we will consider in future research.

4. Please place *** (significance stars) next to the results that are statistically significant in the tables.  

Thank you for this comment, making it more clear which results that are statistically significant. In table 2, a dagger (†) is now used to highlight the significant results. The dagger was chosen as “*” is already used as a footnote in the table.

Reviewer 3 Report

This study analyzes the impact of an organizational level intervention to decrease absences from sickness in public sector employees in Sweden. When looking at net present values, the study finds a positive impact for the approach combining monetary and facilitator support (FSA), while the net present value of monetary support alone (MSA) is negative.

This is a nice contribution and my comments are intended to strengthen and clarify your points.

  • As the authors mention, the selection into the FSA policy intervention group was all but random, since high operational areas with total sickness absence (>10%) and high employee turnover were invited to participate in the program. A possibility is that these were areas with high potential for improvement, while operational areas that selected the MSA approach had smaller potential. It seems that the data for the MSA approach has a large variance of total sickness pre-intervention. Would it be possible to repeat the analysis focusing only on a subset of intervention groups that selected the MSA approach with higher total sickness pre-intervention (for example, focus exclusively on the intervention groups with total sickness pre-intervention above average)? If the results were robust, it would be a further confirmation that the ineffectiveness of the MSA approach is not due to lack of potential, but rather to the mechanism suggested by the authors (i.e. that finding effective interventions that fit workplace challenges is key).

Author Response

Thank you for this positive feedback and for reviewing our manuscript, and the valuable comments on how the manuscript can be improved. Please see answers to your comments below, highlighted in red.

As the authors mention, the selection into the FSA policy intervention group was all but random, since high operational areas with total sickness absence (>10%) and high employee turnover were invited to participate in the program. A possibility is that these were areas with high potential for improvement, while operational areas that selected the MSA approach had smaller potential. It seems that the data for the MSA approach has a large variance of total sickness pre-intervention. Would it be possible to repeat the analysis focusing only on a subset of intervention groups that selected the MSA approach with higher total sickness pre-intervention (for example, focus exclusively on the intervention groups with total sickness pre-intervention above average)? If the results were robust, it would be a further confirmation that the ineffectiveness of the MSA approach is not due to lack of potential, but rather to the mechanism suggested by the authors (i.e. that finding effective interventions that fit workplace challenges is key).

Thank you for this comment on differences in potential for reducing sickness absence between the two approaches. In the discussion, we explain that the intervention groups in the MSA had a higher mean sickness absence compared both to their respective reference group, and the region, which implies a potential to reduce the sickness absence for workplaces participating in this approach as well. In the method section, we have also added information about which sector the participating workplaces came from (mainly patient care, the remaining with service functions within healthcare), where there is a similar distribution between the MSA and the FSA. Besides, the aim of the program was to reduce total sickness absence, suggesting that both the managers and HR applying for funds identified a potential for reducing the sickness absence at their workplace.

Furthermore, since the association between working conditions and sickness absence are both complex and multifactorial, a high sickness absence may not just indicate an increased potential for improvement, but also prolonged and more severe work environment challenges that may require more extensive measures over a longer time to come to terms with these challenges. Thus, it may even be harder to decrease high levels of sickness absence compared to more moderately increased levels of sickness absence within the work environment.

Therefore, the suggestion to separate some of the intervention groups in the MSA have occurred to us, which of course would be possible in theory. However, it would be difficult to decide which intervention groups to separate for such an analysis, as we do not have more knowledge about the characteristics and context for each group. Therefore, there is a risk to introduce systematic differences, affecting the analysis. But we thank the reviewer for the opportunity, and challenge (!) to think this through which has gained us a deeper understanding. We hope that both the Editor and the reviewer will be satisfied with this answer. Please let us know if we misinterpreted the question or if further actions are required.